# AUL is a Better Optimization Metric in PU Learning

## Abstract

Traditional binary classification models are trained and evaluated with fully labeled data which is not common in real life. In non-ideal dataset, only a small fraction of positive data are labeled. Training a model from such partially labeled data is named as positive-unlabeled (PU) learning. A naive solution of PU learning is treating unlabeled samples as negative. However, using biased data, the trained model may converge to non-optimal point and its real performance cannot be well estimated. Recent works try to recover the unbiased result by estimating the proportion of positive samples with mixture proportion estimation (MPE) algorithms, but the model performance is still limited and heavy computational cost is introduced (particularly for big datasets). In this work, we theoretically prove that Area Under Lift curve (AUL) is an unbiased metric in PU learning scenario, and the experimental evaluation on 9 datasets shows that the average absolute error of AUL estimation is only 1/6 of AUC estimation. By experiments we also find that, compared with state-of-the-art AUC-optimization algorithm, AUL-optimization algorithm can not only significantly save the computational cost, but also improve the model performance by up to 10%.

## 1 Introduction

Classic binary classification tasks in machine learning usually assume that all data are fully labeled as positive or negative (PN learning). However, in real-world applications, dataset is usually non-ideal and only a small fraction of positive data are labeled. Training a model from such partially labeled positive data is called positive-unlabeled (PU) learning. Take financial fraud detection as an example. Some fraudulent manners are found and can be labeled as positive, but we cannot simply regard the remaining data as negative, because in most cases only a subset of fraud manners are detected and the remaining data may also contain undetected positive data. As a result, the remaining data can only be regarded as unlabeled. Other typical PU learning applications include text classification, drug discovery, outlier detection, malicious URL detection, online advertise, etc (Yu et al. (2002), Li & Liu (2003), Li et al. (2009), Blanchard et al. (2010), Zhang et al. (2017), Wu et al. (2018)).

A naive way for PU learning is treating unlabeled data as negative and using traditional PN learning algorithms. But the model trained in this way is biased and its prediction results are not reliable (Elkan & Noto (2008)). Some early works try to recover labels for unlabeled data by heuristic algorithms, such as S-EM (Liu et al. (2002)), 1-DNF (Yu et al. (2002)), Rocchio (Li & Liu (2003)), k-means (Chaudhari & Shevade (2012)). But the performance of heuristic algorithms, which is critical to these works, is not guaranteed. Some other kind of methods introduce an unbiased risk estimator to eliminate the bias (Du Plessis et al. (2014), Du Plessis et al. (2015), Kiryo et al. (2017)). However, these methods rely on the knowledge of the proportion of positive samples in unlabeled samples, which is also unknown in practice.

Another annoying problem of PU learning is how to accurately evaluate the model's performance. Model performance is usually evaluated by some metrics, such as accuracy, precision, recall, F-score, AUC (Area Under ROC Curve), etc. During the life cycle of a model, its performance is usually monitored to ensure that the model is keeping a desired level of performance, with the variance and growth of data. In PU learning, the metrics above are also biased due to the lack of proportion of positive samples. Although Menon et al. (2015) proves that the ground-truth AUC (AUC) and

the AUC estimated from PU data ($\text{AUC}^{\text{PU}}$) is linearly correlated, which indicates that $\text{AUC}^{\text{PU}}$ can be used to compare the performances between two models, it's still not possible to evaluate the true performance of a single model. Consider a situation when a model is evaluated on two different PU datasets generated from the same PN dataset but with different positive sample proportions. The ground-truth AUC which indicates the true performance of the model on two datasets are the same, but the $\text{AUC}^{\text{PU}}$ on the two datasets are different. Hence, $\text{AUC}^{\text{PU}}$ cannot be used to directly evaluate the model's performance. Jain et al. (2017) and Ramola et al. (2019) show that they can correct $\text{AUC}^{\text{PU}}$, $\text{accuracy}^{\text{PU}}$, balanced $\text{accuracy}^{\text{PU}}$, $\text{F-score}^{\text{PU}}$ and Matthews correlation coefficient, with the knowledge of proportion of positive samples. However, this proportion is difficult to obtain in practice.

Recently many works focus on *estimating* the proportion of positive samples Du Plessis & Sugiyama (2014), Christoffel et al. (2016), Ramaswamy et al. (2016), Jain et al. (2016), Bekker & Davis (2018), Zeiberg et al. (2020), which are called mixture proportion estimation (MPE) algorithms. Yet according to our experiments on 9 datasets, the estimation methods still introduce some errors and thus make the corrected metrics inaccurate. Besides, the MPE algorithms may also introduce non-trivial computational overhead (by up to 2,000 seconds per proportion estimation in our experiments), which slows down the evaluation process.

In this work, we find that Area Under Lift chart (AUL) (Vuk & Curk (2006), Tufféry (2011)) is a discriminating, unbiased and computation-friendly metric for PU learning. We make the following contributions. a). We theoretically prove that AUL estimation is unbiased to the ground-truth AUL and calculate a theoretical bound of the estimation error. b). We carry out experimental evaluation on 9 datasets and the results show that the average absolute error of AUL estimation is only 1/6 of AUC estimation, which means AUL estimation is more accurate and more stable than AUC estimation. c). By experiments we also find that, compared with state-of-the-art AUC-optimization algorithm, AUL-optimization algorithm can not only significantly save the computational cost, but also improve the model performance by up to 10%.

The remaining of this paper is organized as follows. Section 2 describes the background knowledge. Section 3 theoretically proves the unbiased feature of AUL estimation in PU learning. Section 4 evaluates the performance of AUL estimation by experiments on 9 datasets. Section 5 experimentally shows the performance of AUL-optimization algorithm by applying AUL in PU learning. Section 6 concludes the whole paper.

## 2 BACKGROUND

**Binary Classification Problem:** Let $\mathbb{D} = \{<\boldsymbol{x}_i, y_i>, i = 1, ...n\}$ be a positive and negative (PN) dataset which has $n$ instances. Each tuple $<\boldsymbol{x}_i, y_i>$ is a record, in which $\boldsymbol{x}_i \in \mathbb{R}^d$ is the feature vector and $y_i \in \{1, 0\}$ is the corresponding ground-truth label.

Let $\mathbb{X}^P$, $\mathbb{X}^N$ be the feature vectors set of **p**ositive, **n**egative samples respectively, and $n^P$, $n^N$ be the number of samples in these sets respectively.

$$\mathbb{X}^P = \{\boldsymbol{x}_i | y_i = 1, i = 1, ...n^P\}$$

$$\mathbb{X}^N = \{\boldsymbol{x}_i | y_i = 0, i = 1, ...n^N\}$$

In PU learning, we use $\alpha = \frac{n^P}{n^P + n^N} = \frac{n^P}{n}$ to indicate the proportion of positive samples in all samples.

**Confusion Matrix:** A confusion matrix is used to discriminate the model performance of different binary classification algorithms. In confusion matrix, true positive (TP) (actual label and predicted label are both positive), true negative (TN) (actual label and predicted label are both negative), false positive (FP) (actually negative but predicted as positive), and false negative (FN) (actually positive but predicted as negative) are counted according to model's outputs. Obviously, $n^{TP} + n^{FN} = n^P$, $n^{TN} + n^{FP} = n^N$.

**ROC:** Since the numbers of TP, TN, FP and FN in a confusion matrix are highly related to the classification threshold ($\theta$), Receiver Operating Characteristic (ROC) curve (Fawcett & Tom, 2003) is proposed to plot $(x, y) = (fpr(\theta), tpr(\theta))$ over all possible classification thresholds $\theta$. In some literature, $tpr$ is also known as sensitivity and the value of $1 - fpr$ is called specificity.

$$\text{true positive rate } (tpr) = \frac{n^{TP}}{n^P}, \text{ false positive rate } (fpr) = \frac{n^{FN}}{n^N}$$

**AUC:** As a curve, ROC is not convenient enough to describe the model performance. Consequently, the Area Under ROC Curve (AUC), which is a single value, is proposed and widely used as a metric to evaluate a binary classification algorithm. AUC provides a summary of model performance under all possible classification thresholds. It also provides an elegant probabilistic interpretation that AUC is the probability of correct ranking between a random positive sample and a random negative sample(Hanley & McNeil, 1982), which is a kind of ranking capability. According to Vuk & Curk (2006), for a model $g : \mathbb{R}^d \to \mathbb{R}$, AUC can be computed as follows,

$$AUC = \frac{1}{n^P n^N} \sum_{\boldsymbol{x}_i \in \mathbb{X}^P} \sum_{\boldsymbol{x}_j \in \mathbb{X}^N} S\left(g(\boldsymbol{x}_i), g(\boldsymbol{x}_j)\right) \tag{1}$$

where

$$S(a, b) = \begin{cases} 1 & a > b \\ \frac{1}{2} & a = b \\ 0 & a < b \end{cases}$$

It is worth noting that, there are other ways to calculate AUC, but they are essentially the same.

**AUL:** Lift curve, which is popular in econometrics to decide a suitable marketing strategy (Tufféry (2011), Vuk & Curk (2006)), has not been well studied in machine learning field. Lift curve can be seen as a variant of ROC and it illustrates $(x, y) = (Y_{rate}(\theta), tpr(\theta))$ over all possible classification thresholds $\theta$. $Y_{rate}$ represents the proportion of samples predicted as positive.

$$Y_{rate} = \frac{n^{TP} + n^{FP}}{n}$$

In the curve figure, Lift curve has the same y-axis as ROC curve, but a different x-axis.

Area Under Lift chart (AUL) (Vuk & Curk (2006), Tufféry (2011)), can also be used as a metric to evaluate the model performance. One way to compute AUL is

$$AUL = \frac{1}{n^P n} \sum_{\boldsymbol{x}_i \in \mathbb{X}^P} \sum_{\boldsymbol{x}_j \in \mathbb{X}^P \cup \mathbb{X}^N} S\left(g(\boldsymbol{x}_i), g(\boldsymbol{x}_j)\right) \tag{2}$$

Essentially, AUL can be regarded as the probability of correct ranking between a random positive sample and a random sample. $AUL$ and $AUC$ is linearly related (Tufféry, 2011), i.e.

$$AUL = 0.5\alpha + (1 - \alpha)AUC$$

which shows that AUL has the same discriminating power with AUC.

## 3 UNBIASEDNESS OF AUL ESTIMATION IN PU LEARNING: THEORETICAL PROOF

A PU dataset $\mathbb{D}' = \{<\boldsymbol{x}_i, y_i, s_i>, s_i \in \{1, 0\}, i = 1, ...n\}$ is generated by sampling a subset of positive data as labeled and leaving remain as unlabeled from $\mathbb{D}$. In $\mathbb{D}'$, $s_i$ is the observed label and $y_i$ is the ground-truth label which may be unknown. If $s_i = 1$, we can confirm $y_i = 1$ (positive). If $s_i = 0$, $y_i$ would be 1 or 0. In this paper, we assume that the labeled data is Select Completely At Random (SCAR) (Bekker & Davis, 2018) from positive data. Therefore the distribution of labeled samples in $\mathbb{D}'$ are the same as the distribution of positive samples in $\mathbb{D}$. Let $\mathbb{X}^L$, $\mathbb{X}^U$ be the feature vectors set of **l**abeled and **u**nlabeled samples respectively, and $n^L$, $n^U$ be the number of samples in these sets respectively.

$$\mathbb{X}^L = \{\boldsymbol{x}_i | s_i = 1, i = 1, ...n^L\}$$
$$\mathbb{X}^U = \{\boldsymbol{x}_i | s_i = 0, i = 1, ...n^U\}$$

We use $\beta = \frac{n^L}{n^P}$ to indicate the proportion of labeled samples in positive samples.

**AUC Estimation is Biased**

To calculate AUC with PU dataset ($AUC^{PU}$), unlabeled data is regarded as negative, thus we have

$$AUC^{PU} = \frac{1}{n^L n^U} \sum_{\boldsymbol{x}_i \in \mathbb{X}^L} \sum_{\boldsymbol{x}_j \in \mathbb{X}^U} S\left(g(\boldsymbol{x}_i), g(\boldsymbol{x}_j)\right) \tag{3}$$

where function $S$ is the same as in Eq.1. The expectation of $AUC^{PU}$ over the distribution of $\mathbb{D}'$ is

$$\mathbb{E}[AUC^{PU}] = \frac{1-\alpha}{1-\alpha\beta}(AUC - 0.5) + 0.5$$

This formula is slightly different from the one in Menon et al. (2015). Here we define AUC on a specific dataset but not on a distribution. This formula indicates that $AUC^{PU}$ is an biased estimation of $AUC$. We demonstrate the bias on an example dataset (1a), which contains 20 samples sorted by prediction score. Figure 1b illustrates two ROC curves on this dataset. *curve-ROC* is ploted with ground-truth label $y$ and *curve-ROC$^{PU}$* is ploted with observed label $s$. We can see that *curve-ROC* is almost above *curve-ROC$^{PU}$*. The corresponding AUC is $AUC = 0.740$ and $AUC^{PU} = 0.653$ respectively. There is a big difference (0.087) among the two measurements.

As we discussed in section 1, (Jain et al., 2017) tries to recover $AUC$ from $AUC^{PU}$ from the estimation of $\frac{1-\alpha}{1-\alpha\beta}$. To estimate $\frac{1-\alpha}{1-\alpha\beta}$, some works(Elkan & Noto (2008), Du Plessis & Sugiyama (2014), Sanderson & Scott (2014), Jain et al. (2016), Ramaswamy et al. (2016),Christoffel et al. (2016), Bekker & Davis (2018), Zeiberg et al. (2020)) develop their mixture proportion estimation (MPE) algorithms. But according to our experiment on 9 datasats, these algorithms are neither accurate enough nor time saving.

**AUL Estimation is Unbiased**

Similar to $AUC^{PU}$, AUL with PU dataset ($AUL^{PU}$) can be calculated as

$$AUL^{PU} = \frac{1}{n^L n} \sum_{\boldsymbol{x}_i \in \mathbb{X}^L} \sum_{\boldsymbol{x}_j \in \mathbb{X}^L \cup \mathbb{X}^U} S\left(g(\boldsymbol{x}_i), g(\boldsymbol{x}_j)\right) \tag{4}$$

Unlike $AUC^{PU}$, $AUL^{PU}$ is unbiased estimation of $AUL$. In contrast to Figure 1b, Figure 1c illustrates two lift curves which are very close to each other. *curve-lift* is ploted with ground-truth label $y$ and *curve-lift$^{PU}$* is ploted with observed label $s$. The corresponding AUL, $AUL = 0.620$ and $AUL^{PU} = 0.615$, are very close. We then prove the unbiasedness.

**Theorem 1** *For a given classifier $g : \mathbb{R}^d \to \mathbb{R}$, a PN dataset $\mathbb{D}$ with the proportion of labeled samples in positive samples $\beta = \frac{n^L}{n^P}$, a PU dataset $\mathbb{D}'$ can be generated following SCAR, the expectation and variance of $AUL^{PU}$ over the distribution of $\mathbb{D}'$ are as follows,*

$$\mathbb{E}[AUL^{PU}] = AUL$$

$$\mathbb{V}\mathrm{ar}[AUL^{PU}] = \frac{n^P - n^L}{n^P - 1}\frac{\sigma^2}{n^L}$$

*where $\sigma^2$ is the variance of $\left\{ \frac{1}{n}\sum_{\boldsymbol{x}_j \in \mathbb{X}^P \cup \mathbb{X}^N} S\left(g(\boldsymbol{x}_i), g(\boldsymbol{x}_j)\right), i = 1, ...n^P \right\}$.*

**Proof** *Let*

$$t_{\boldsymbol{x}_i} = \frac{1}{n} \sum_{\boldsymbol{x}_j \in \mathbb{X}^P \cup \mathbb{X}^N} S\left(g(\boldsymbol{x}_i), g(\boldsymbol{x}_j)\right) = \frac{1}{n} \sum_{\boldsymbol{x}_j \in \mathbb{X}^L \cup \mathbb{X}^U} S\left(g(\boldsymbol{x}_i), g(\boldsymbol{x}_j)\right)$$

*then,*

$$AUL = \frac{1}{n^P} \sum_{\boldsymbol{x}_i \in \mathbb{X}^P} t_{\boldsymbol{x}_i}$$

$$AUL^{PU} = \frac{1}{n^L} \sum_{\boldsymbol{x}_i \in \mathbb{X}^L} t_{\boldsymbol{x}_i}$$

$\mathbb{X}^L$ *is generated by random sampling without replacement from $\mathbb{X}^P$, hence $AUL^{PU}$ is the estimation of the mean of $\{t_{\boldsymbol{x}_i}, \boldsymbol{x}_i \in \mathbb{X}^P\}$ which is $AUL$. According the theory of simple random sampling*

*without replacement (Lohr (2009)), the estimated population mean $AUL^{PU}$ is an unbiased estimator of the population mean $AUL$, i.e. $\mathbb{E}[AUL^{PU}] = AUL$, and variance of $AUL^{PU}$ is*

$$\mathbb{V}\text{ar}[AUL^{PU}] = (1 - \frac{n^L}{n^P})\frac{1}{n^L}\left(\frac{\sum_{\boldsymbol{x}_i \in \mathbb{X}^P}(t_{\boldsymbol{x}_i} - \bar{t})^2}{n^P - 1}\right)$$

$$= \frac{n^P - n^L}{n^P - 1}\frac{1}{n^L}\left(\frac{\sum_{\boldsymbol{x}_i \in \mathbb{X}^P}(t_{\boldsymbol{x}_i} - \bar{t})^2}{n^P}\right)$$

$$= \frac{n^P - n^L}{n^P - 1}\frac{\sigma^2}{n^L}$$

According to Theorem 1, applying Chebyshev's inequality, we have

$$P\left(|AUL - AUL^{PU}| \geq \epsilon\right) \leq \frac{\text{Var}}{\epsilon} = \frac{\sigma^2}{n^L \epsilon}(1 - \beta)\frac{n^P}{n^P - 1}$$

where Var is the variance of $AUL^{PU}$, note that $0 < t_{\boldsymbol{x}_i} < 1$, hence

$$\sigma^2 = \mathbb{E}\left[(t - \bar{t})^2\right]$$

$$\leq \mathbb{E}\left[(t - \bar{t})^2 - (0 - \bar{t})(\bar{t} - 1)\right]$$

$$= \mathbb{E}\left[-2t\bar{t} + \bar{t}^2 + t\right]$$

$$= \bar{t} - \bar{t}^2 \leq \frac{1}{4}$$

then

$$P\left(|AUL - AUL^{PU}| \geq \epsilon\right) \leq \frac{1 - \beta}{4n^L \epsilon}\frac{n^P}{n^P - 1} \approx \frac{1 - \beta}{4n^L \epsilon}$$

This equation gives a theoretical bound for the error between $AUL$ and $AUL^{PU}$. Hundreds of labeled samples $n^L$ can reduce the error to an acceptable level.

| score | 0.92 | 0.82 | 0.73 | 0.66 | 0.6 | 0.58 | 0.54 | 0.5 | 0.45 | 0.43 |
|---|---|---|---|---|---|---|---|---|---|---|
| $y$ | 1 | 1 | 0 | 1 | 1 | 1 | 1 | 0 | 0 | 0 |
| $s$ | 1 | 0 | 0 | 1 | 0 | 0 | 1 | 0 | 0 | 0 |

| score | 0.41 | 0.39 | 0.38 | 0.36 | 0.35 | 0.3 | 0.25 | 0.2 | 0.15 | 0.1 |
|---|---|---|---|---|---|---|---|---|---|---|
| $y$ | 1 | 0 | 1 | 1 | 0 | 0 | 0 | 1 | 0 | 0 |
| $s$ | 1 | 0 | 0 | 0 | 0 | 0 | 0 | 1 | 0 | 0 |

(a)

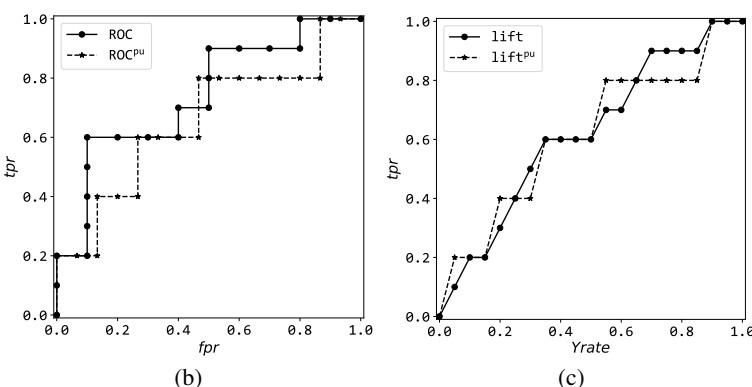

(b)      (c)

Figure 1: AUC and AUL estimation on an example dataset 1a (20 instances, $\alpha = 0.5, \beta = 0.5$). Each instance has a prediction score (*score*) given by a certain model, a ground-truth label $y$ and an observed label $s$.

# 4  EXPERIMENTAL EVALUATION OF AUL ESTIMATION

## 4.1  DATASETS

The experiment involves 9 real-life datasets from UCI Machining Learning Repository (Dua & Graff, 2017), which are listed in table 1. To create binary classification datasets we do a little modification on the original datasets' 'target'. If it's a regression dataset we take a proper threshold and transfer it to a binary classification dataset. For a multi-class dataset, we chose one class as positive and the remaining as negative. We also transfer categorical features to numerical features using one-hot encoding. Considering the computing overhead of MPE algorithm, we limit the size of PN dataset around 4,000. In order to generate PU dataset, we use random sampling without replacement method to select a subset of positive samples as labeled data and the remaining are regarded as unlabeled data. There are three settings of $\beta = \{0.1, 0.2, 0.4\}$. For each dataset setting, 50 PU datasets are generated.

## 4.2  COMPARISON AMONG AUC&AUL ESTIMATION METHODS

High estimating accuracy and low cost are two important indexes for AUC/AUL estimation. Accuracy indicates how close the estimated AUC/AUL is to the ground-truth AUC/AUL. Cost indicates how much time (computation power) an estimation process takes. AUC and AUL estimation rely on a model's output. Therefore, for each dataset, we train a classifier using lightGBM (Ke et al., 2017) to simulate the classifier to be evaluated. The performance of this classifier is rational, not too bad nor totally perfect.

To compare accuracy, we firstly compute the ground-truth value of AUC ($AUC$) and AUL ($AUL$) on fully labeled PN dataset. Then we calculate the estimated AUC ($AUC^{est}$) and AUL ($AUL^{PU}$) on PU dataset and compare them with $AUC$ and $AUL$. To calculate $AUC^{est}$, we use the direct conversion method following Jain et al. (2017). This method firstly obtains $AUC^{PU}$ which is calculated by treating unlabeled data as negative and then estimates $AUC^{est}$ with it. The estimating step needs a mixture proportion estimation (MPE) algorithm to estimate the proportion of positive samples in unlabeled samples. Three MPE algorithms are used in this experiment. Ramaswamy et al. (2016) provides two algorithms named as **KM1** and **KM2**. Zeiberg et al. (2020) which is based on distance curve is named as **Distance**. It's worth noting that Zeiberg et al. (2020) has compare their solution with all the existing MPE algorithms and claim themselves as the best so far. The corresponding estimated AUC of the three algorithms are $AUC^{est}_{KM1}$, $AUC^{est}_{KM2}$ and $AUC^{est}_{Distance}$ respectively. We use the codes [1] [2] provided by the two papers.

Table 1 shows the Mean Absolutely Error (MAE) results for each dataset. $AUC$ and $AUL$ are ground-truth value computed on fully labeled datasets. The estimation processes run 50 times for each dataset setting and get $AUC^{est}_{KM1}$, $AUC^{est}_{KM2}$, $AUC^{est}_{Distance}$ and $AUL^{PU}$. $MAE^{KM1}_{AUC}$, $MAE^{KM2}_{AUC}$, $MAE^{Distance}_{AUC}$, $MAE_{AUL}$ are the mean values of $|AUC^{est}_{KM1} - AUC|$, $|AUC^{est}_{KM2} - AUC|$, $|AUC^{est}_{Distance} - AUC|$, $|AUL^{PU} - AUL|$ respectively. In all settings, MAE of AUL estimation outperforms AUC estimation. The average MAE of AUL estimation on all 3*9 dataset settings is only 1/6 of the best AUC estimation method(Distance).

Figure 2 illustrates the error ($AUC^{est} - AUC$ or $AUL^{PU} - AUL$) distributions of 150 estimation results (3 $\beta$ settings for each dataset, each setting run estimation for 50 times) per dataset using boxplot. Each boxplot plots the minimum, the first quartile, the sample median, third quartile and the maximum value in ascending order by five horizontal lines. The mean value is ploted with a black diamond. This figure shows that the mean value of AUL estimation error is the closest to 0 and the interquartile range (range between first and third quartile) of AUL estimation error is the smallest. It indicates that AUL estimation is more accurate and stable than AUC estimation. We also noticed that when $\beta$ becomes larger, the interquartile range gets smaller. This is consistent with our conclusion in section 3.

---

[1] http://web.eecs.umich.edu/~cscott/code.html#kmpe
[2] https://github.com/Dzeiberg/ClassPriorEstimation

Table 1: Mean Absolutely Error (MAE) of AUC/AUL estimation methods on 9 datasets.

| Dataset | #n | $\alpha$ | $\beta$ | AUC | AUL | $MAE_{AUC}^{KM1}$ | $MAE_{AUC}^{KM2}$ | $MAE_{AUC}^{Distance}$ | $MAE_{AUL}$ |
|---|---|---|---|---|---|---|---|---|---|
| Abalone | 4177 | 0.498 | 0.1 | 0.807 | 0.654 | 0.065 | 0.035 | 0.037 | 0.014 |
| Abalone | 4177 | 0.498 | 0.2 | 0.807 | 0.654 | 0.043 | 0.043 | 0.031 | 0.008 |
| Abalone | 4177 | 0.498 | 0.4 | 0.807 | 0.654 | 0.027 | 0.054 | 0.034 | 0.005 |
| Airfoil | 1503 | 0.483 | 0.1 | 0.845 | 0.678 | 0.110 | 0.088 | 0.093 | 0.019 |
| Airfoil | 1503 | 0.483 | 0.2 | 0.845 | 0.678 | 0.087 | 0.048 | 0.064 | 0.012 |
| Airfoil | 1503 | 0.483 | 0.4 | 0.845 | 0.678 | 0.057 | 0.026 | 0.041 | 0.009 |
| Anuran | 4000 | 0.614 | 0.1 | 0.887 | 0.649 | 0.058 | 0.070 | 0.073 | 0.011 |
| Anuran | 4000 | 0.614 | 0.2 | 0.887 | 0.649 | 0.054 | 0.040 | 0.065 | 0.007 |
| Anuran | 4000 | 0.614 | 0.4 | 0.887 | 0.649 | 0.055 | 0.024 | 0.026 | 0.004 |
| Concrete | 1030 | 0.472 | 0.1 | 0.875 | 0.698 | 0.091 | 0.113 | 0.100 | 0.022 |
| Concrete | 1030 | 0.472 | 0.2 | 0.875 | 0.698 | 0.096 | 0.066 | 0.061 | 0.016 |
| Concrete | 1030 | 0.472 | 0.4 | 0.875 | 0.698 | 0.105 | 0.041 | 0.083 | 0.009 |
| Landsat | 4435 | 0.242 | 0.1 | 0.756 | 0.694 | 0.018 | 0.018 | 0.049 | 0.015 |
| Landsat | 4435 | 0.242 | 0.2 | 0.756 | 0.694 | 0.014 | 0.012 | 0.030 | 0.011 |
| Landsat | 4435 | 0.242 | 0.4 | 0.756 | 0.694 | 0.010 | 0.008 | 0.010 | 0.007 |
| Mushroom | 4000 | 0.518 | 0.1 | 0.942 | 0.713 | 0.028 | 0.066 | 0.042 | 0.011 |
| Mushroom | 4000 | 0.518 | 0.2 | 0.942 | 0.713 | 0.021 | 0.028 | 0.030 | 0.006 |
| Mushroom | 4000 | 0.518 | 0.4 | 0.942 | 0.713 | 0.027 | 0.018 | 0.026 | 0.004 |
| Pageblock | 4000 | 0.898 | 0.1 | 0.891 | 0.54 | 0.177 | 0.300 | 0.201 | 0.011 |
| Pageblock | 4000 | 0.898 | 0.2 | 0.891 | 0.54 | 0.110 | 0.249 | 0.183 | 0.008 |
| Pageblock | 4000 | 0.898 | 0.4 | 0.891 | 0.54 | 0.111 | 0.159 | 0.163 | 0.005 |
| Spambase | 4601 | 0.394 | 0.1 | 0.841 | 0.706 | 0.111 | 0.068 | 0.037 | 0.014 |
| Spambase | 4601 | 0.394 | 0.2 | 0.841 | 0.706 | 0.084 | 0.106 | 0.018 | 0.008 |
| Spambase | 4601 | 0.394 | 0.4 | 0.841 | 0.706 | 0.056 | 0.131 | 0.008 | 0.005 |
| Waveform | 4000 | 0.329 | 0.1 | 0.926 | 0.785 | 0.074 | 0.073 | 0.105 | 0.009 |
| Waveform | 4000 | 0.329 | 0.2 | 0.926 | 0.785 | 0.073 | 0.074 | 0.025 | 0.005 |
| Waveform | 4000 | 0.329 | 0.4 | 0.926 | 0.785 | 0.064 | 0.074 | 0.028 | 0.004 |

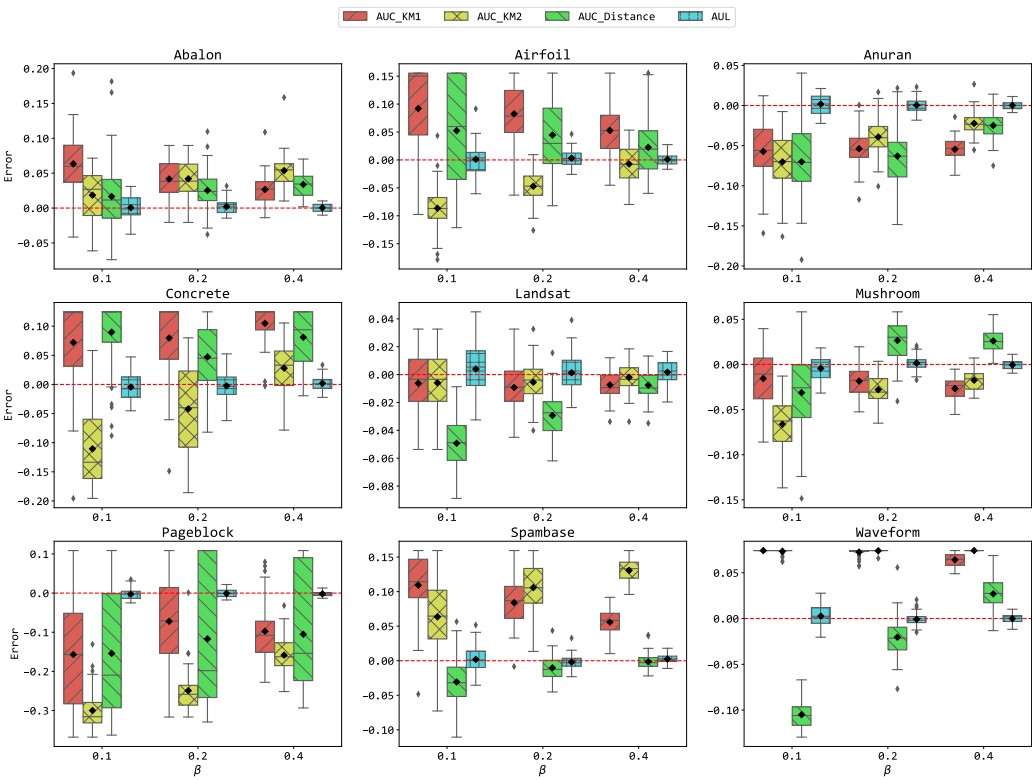

Figure 2: Estimation error distribution on 9 datasets.

The cost of AUC estimation includes $AUC^{PU}$ calculation cost and MPE algorithm's calculation cost. The cost of AUL estimation includes $AUL^{PU}$ calculation cost only, which is nearly the same as $AUC^{PU}$ calculation cost and it is negligible (less than 1 second). However, the MPE algorithms cost is much larger. We conducted an experiment to count the time cost of MPE algorithms. For a dataset with 8,000 instances and 117 features, the time cost of Ramaswamy et al. (2016)'s method grows fast when the dataset's size grows (4 seconds for 1,000 samples, 2,000 seconds for 8,000 samples). Zeiberg et al. (2020) introduces an univariate transform to reduce the dimensionality of data. Therefor, their MPE method get a significant acceleration (16 seconds for 8,000 samples). But the selection of univariate transform is another time consuming problem. A careless selection of the transform will make the estimating result inaccurate. In the work of Zeiberg et al. (2020), a lot of effort was made to select the optimal transform for each PU dataset.

## 5    EXPERIMENTAL EVALUATION OF AUL-OPTIMIZATION ALGORITHM

Optimizing AUC is a direct and popular way for training binary classifier models. Sakai et al. (2018) develops a AUC-optimization algorithm named as PU_AUC in PU learning scenario. Following this idea we also implement an AUL-optimization algorithm named as PU_AUL in a similar way of PU_AUC. PU_AUC and PU_AUL share the same Gaussian kernel basis function, which is adopted by Sakai et al. (2018). The datasets we used in this section are the same with that in section 4 excepting $\beta$ is fixed at 0.1. Because PU_AUC algorithm requires the proportion of positive samples estimated by MPE algorithms, we choose the best performed MPE algorithm for PU_AUC in each dataset. The metric used for model performance comparison is AUC which is calculated on ground-truth PN data.

Figure 3 shows that PU_AUL achieves better performance on 8/9 datasets (average 2.5% improvement). Because PU_AUC may suffer from the error of MPE algorithm, on 'Concrete' dataset, PU_AUL outperforms PU_AUC by 10%. Because MPE calculation is not required in PU_AUL, it runs much faster.

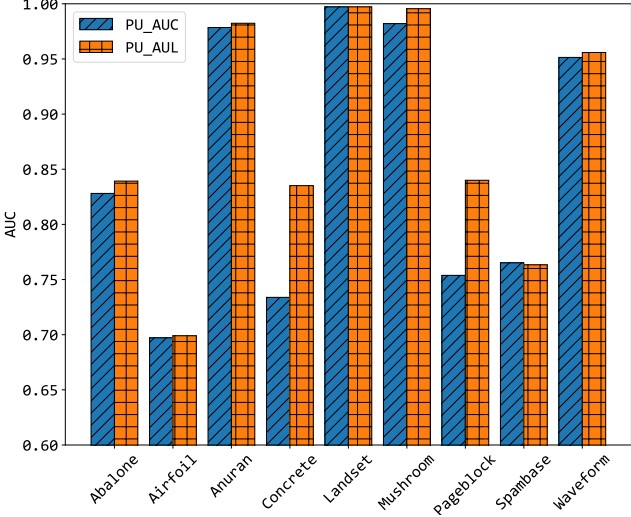

Figure 3: PU_AUL outperforms PU_AUC by up to 10% on 9 datasets.

## 6    CONCLUSION

In this paper, we suggest replacing AUC by AUL for both model evaluation and model training in PU learning scenario. Comparing with AUC, AUL is an unbiased metric and can be computed efficiently. Existing MPE algorithms, which is a necessary for AUC-optimization algorithms, have been proved to be inaccurate and high cost. Besides, choosing a good set of parameters for MPE algorithms in order to get a good estimation result may even take more time.

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
