# OpenReview forum: "AUL is a better optimization metric  in PU learning"
_ICLR.cc/2021/Conference — Reject_

### Official Review · AnonReviewer4 · 2020-10-16
**An interesting proposition but work is too premature**

**Rating:** 3
**Confidence:** 3

**Review:**

This paper examines the positive-unlabeled (PU) learning setting, and recommends the usage of the area under the lift curve, or AUL, as an unbiased estimate of the AUL under the fully labeled setting. This justifies proposing to use an AUL-optimization algorithm to train binary classifiers.

Although I am not entirely sure this is fitting for a conference focused on representation learning, this is a question of interest in machine learning at large.

I think this is an interesting proposition, supported by the experimental results. However, I do have a number of concerns that make me think this work is too premature and I lean towards rejection.

1) I do not understand where the bias of $AUC^{PU}$ comes from in the equation below equation (3). Indeed, it is not obvious how it is derived from (3), and Jain et al. (2017) obtain
$$
   (\beta-\alpha) (AUC - \frac12) + \frac12.
$$

2) I disagree with the bound on $P(|AUL - AUL^{PU}| \geq \epsilon)$. Indeed, applying Chebychev's inequality, I think this should be upper bounded by $\frac{\text{Var}}{\epsilon^2}$ and not $\frac{\text{Var}}{\epsilon}$. Hence the final bound should be
$$
    P(|AUL - AUL^{PU}| \geq \epsilon) \leq \frac{1 - \beta}{4 n^L \epsilon^2}.
$$
This is unfortunate, because taking $\epsilon = 1\%$, and for example $\beta = 0.5$,
$$
   \frac{1-\beta}{4 \epsilon^2} = 1250,
$$
so the sentence "Hundreds of labeled samples can reduce the error to an acceptable level" does not hold any longer -- it's now "tens of thousands of samples".
On the numerical experiments of Table 1, you can indeed check that for the Airfol data set, with $\beta = 0.1$,
$$
    \frac{1-\beta}{4 n^L \epsilon^2} = \frac{1-\beta}{4 \alpha \beta n \epsilon^2} \approx \frac{1.8 \times 10^{-3}}{\epsilon^2},
$$
so $P(|AUL - AUL^{PU}| \geq 10^{-2})$ can only be bounded by 18, and indeed $MAE_{AUL}=0.019$.

3) Although some of the papers that are cited also use UCI data sets, I am wondering whether the values of $\alpha$ that are being tested are realistic. Indeed, it is my impression that PU settings occur mostly when the relative proportion of positive examples in the data is really small (fraud or outlier detection, malicious URL detection, drug discovery), which is why it makes sense to treat the unlabeled as negative, but here $\alpha$ is between 0.3 and 0.9.

5) The AUL-optimization algorithm is an interesting proposal, and I think the paper would greatly benefit from having more details about this algorithm, even if it is "only" an adaptation of the algorithm of Sakai et al. (2018).

6) On Figure 3, while for some of the data sets (Pageblock, Concrete) it is obvious that PU_AUL is performing much better than PU_AUC, it is not clear for all of them (in particular Landset, Anuran, Abalone, Airfoil) whether the difference in performance is significative. I would recommand plotting error bars on multiple cross-validation runs and/or computing statistical tests.

In addition, I have several minor comments.

1) The work is conducted under the assumption that labeled examples are selected completely at random among the positives and I think this should be mentioned explicitly in the abstract and Introduction. Indeed, this assumption may not hold in practice, for example in biological applications (the molecules or interactions labeled as positives are those that have been biologically investigated, and these investigations did not occur uniformly at random).

2) I don't think the proof that $\sigma^2$ is less than 1/4 is correct. The result is known as Popoviciu's inequality on variances, and if you want to prove it you should use
$$
   \mathbb{E}[(t - \bar{t})^2] \leq \mathbb{E}[(t - \bar{t})^2 - t(t-1)],
$$
which is correct because $t(t-1) < 0$. The equation that is given in the paper,
$$
   \mathbb{E}[(t - \bar{t})^2] \leq \mathbb{E}[(t - \bar{t})^2 - (0-\bar{t})(\bar{t}-1)],
$$
is not true because $(0-\bar{t})(\bar{t}-1) > 0$.

3) There are a number of typos in the text. "online advertise", "Select Completely at Random", etc.
The definition of fpr in Section 2, paragraph "ROC", should be
$$
   \frac{n^{FP}}{n^N}.
$$

4) In Section 4.2, I find it odd to compare $|AUC^{est}-AUC|$ to $|AUL^{PU}-AUL|$, because $AUC$ and $AUL$ don't take the same values. I think these two quantities should be normalized by $AUC$ and $AUL$ respectively, reporting $|AUC^{est}-AUC|/AUC$ and $|AUL^{PU}-AUL|/AUL.$ The same holds, of course, for $AUC^{KM1}$ and $AUC^{KM2}$.
I note that it does not seem to change the conclusions that can be drawn from Table 1.

5) In biological and drug discovery applications, it has become rather mainstream to evaluate the performance of PU-learning algorithms using the cumulative distribution function of the ranking of positive samples among all test samples in a leave-one-out setting as in Mordelet and Vert (2011). I think it would be warranted to discuss this method as well.
Another relevant reference seems to be Jiang et al. (2020).

References
Mordelet and Vert, BMC Bioinformatics 2011, 12:389 http://www.biomedcentral.com/1471-2105/12/389
Liwei Jiang, Dan Li, Qisheng Wang, Shuai Wang, Songtao Wang. Improving Positive Unlabeled Learning: Practical AUL Estimation and New Training Method for Extremely Imbalanced Data Sets.  https://arxiv.org/abs/2004.09820

---

### Official Review · AnonReviewer1 · 2020-11-05
**new metric for model evaluation but lacking on methodology and potential impact**

**Rating:** 5
**Confidence:** 3

**Review:**


The paper argues that AUL is a better metric than AUC under the PU (positive and unlabeled data) learning setup in the sense that it leads to an unbiased estimator in this setting, which is not the case for the commonly used and known metric - AUC. It is also argued that it leads to better performance than those methods which directly optimize an AUC based metric and computationally efficient to evaluate than methods which attempt to estimate the unknown parameters (\alpha, \beta in the paper). The appropriateness of this setting in the PU learning setting is demonstrated on the UCI datasets.

The main contribution of the paper is to bring in the AUL metric in the context of PU learning, which seems rather new from the ML perspective. However, there are following concerns regarding the methodology and experimentation :

- The paper repeatedly makes the claim that existing work, including recent ones, which work on estimation of \frac{1-\alpha}{1-\alpha\beta} do not work well on their setup, without discussing why that happens. It is not discussed if there is something wrong in these papers. In my opinion, one needs to give reasonable arguments why these methods do not work, instead of simply saying these did not work.

- The proof of Theorem 1 is not quite clear, and in particular, how equality defining the t_{x_i} holds. Is it related to the SCAR assumption made in the paper. The proof needs to be more details, and it should be clarified where the SCAR assumption is used.

- The SCAR assumption is vaguely defined in words. In the context of the paper, it should be formally defined in terms of quantities already metioned such as \alpha and \beta. Also, is the assumption not too strong, and do other papers make this assumption. How to verify that this assumption holds in practice.

- The paper argues about the computational advantage of their method compared to other methods. However, it is evaluated on small scale UCI datasets,

- There seem to be incorrect usage of PU/PN learning in various places : (i) When defining \alpha in section 2, the sentence says "In PU learning, we ...", should it not be "In PN learning, we ...", (ii) In the statement of Theorem 1, "a PN dataset \mathbb{D} with the proportion of labeled samples in positive samples \beta= ..." - does it make sense to talk about labeled and unlabeled data when talking about PN dataset.

- Comparison with a recent related work is missing - Class Prior Estimation with Biased Positives and Unlabeled Examples - AAAI 2020

---

### Official Review · AnonReviewer6 · 2020-11-07
**a new PU learning optimization metric**

**Rating:** 5
**Confidence:** 4

**Review:**

In this paper, the author proposed to use Area Under Lift chart (AUL) as a new optimization metric for positive unlabeled (PU) learning. The proposed AUL can be estimated unbiasedly from PU data, without the need to estimate the mixture proportions. Experiments on several datasets show that the proposed method outperforms AUC optimization algorithms.

It seems that the authors consider AUC as the evaluation metric of PU learning algorithms. Why not use the classification error to evaluate the learned models? For example, after the mixture proportions are estimated, a simple importance reweighting method could be used to train a classifier.

From the theoretical analysis in the paper, we can only see that the AUL_PU is an unbiased estimator of AUL. However, the link between AUC on the classifier learned by optimizing AUL and the classifier learned directly by AUC is not clear. The authors claim that this might be due to the error in mixture proportion estimation. Does this imply that AUL_PU has no advantages if the mixture proportion is correctly estimated? The author could do some ablation study, for example, training AUC by using ground-truth mixture proportion,  to further analyze the accuracy of the classifier trained by AUC and AUL.

---

### Decision · Program_Chairs · 2021-01-07
**Final Decision**

**Decision:**

Reject

**Comment:**

This paper proposes a new metric, AUL, for classification problems with unbalanced data. The paper proves that it is unbiased and it does not need to estimate the mixture proportions, a traditional approach. Empirical results show improvement.
While the results are novel, the referees pointed out several concerns
1. Positioning of the results in the context of existing work. It would be helpful if the paper can establish theoretical links to existing metrics such as  AUC,
make a very precise statement about what is lacking in those approaches
2. Some of the results, may have bugs in the proof.
3. More ablation studies are needed

In summary the paper has good ideas but maybe premature for publication.